# Identification of Multiple Diseases in Apple Leaf Based on Optimized Lightweight Convolutional Neural Network

**DOI:** 10.3390/plants13111535

**Published:** 2024-06-01

**Authors:** Bin Wang, Hua Yang, Shujuan Zhang, Lili Li

**Affiliations:** 1College of Information Science and Engineering, Shanxi Agricultural University, Jinzhong 030801, China; wangbin1759@126.com (B.W.); yanghuaxky@126.com (H.Y.); 2College of Agricultural Engineering, Shanxi Agricultural University, Jinzhong 030801, China; sxndzsj123@163.com

**Keywords:** apple leaf disease, complex environment, multiple diseases, optimized RegNet

## Abstract

In this study, our aim is to find an effective method to solve the problem of disease similarity caused by multiple diseases occurring on the same leaf. This study proposes the use of an optimized RegNet model to identify seven common apple leaf diseases. We conducted comparisons and analyses on the impact of various factors, such as training methods, data expansion methods, optimizer selection, image background, and other factors, on model performance. The findings suggest that utilizing offline expansion and transfer learning to fine-tune all layer parameters can enhance the model’s classification performance, while complex image backgrounds significantly influence model performance. Additionally, the optimized RegNet network model demonstrates good generalization ability for both datasets, achieving testing accuracies of 93.85% and 99.23%, respectively. These results highlight the potential of the optimized RegNet network model to achieve high-precision identification of different diseases on the same apple leaf under complex field backgrounds. This will be of great significance for intelligent disease identification in apple orchards in the future.

## 1. Introduction

China is the world’s largest producer of fruit, with the largest crop area and output of apples in the world. As one of the most widely consumed fruits in the country, apples play a vital role in economic development and people’s daily lives. However, during the growth of apples, a variety of diseases, such as apple rust, early defoliation, and scab, can occur on the apple leaf area due to natural environmental factors. These diseases seriously hinder the normal growth of apples, affect their yield and quality, and cause significant economic losses to the apple industry. At present, identifying the type of apple leaf diseases requires a manual and experienced approach. This results in low accuracy, is time-consuming and laborious, and results in the spraying of large amounts of pesticide and a low utilization rate, making it difficult to meet the needs of large-scale production.

Due to the complex symptoms of apple leaf diseases, the wrong assessment will lead to the overuse of pesticides, which will not only fail to prevent and control the disease but also lead to a decline in yield and quality, causing environmental problems. Therefore, the automatic identification of apple leaf diseases can provide an effective reference for apple disease control. Therefore, it is of great significance to realize the intelligent, rapid, and accurate identification of apple leaf diseases.

In recent years, many scholars have attempted to use machine learning techniques to design apple leaf disease recognition algorithms. For example, the feature vector is established by using information about color, shape, and texture, then constructing the model for disease recognition [1,2]. For example, Wang et al. [3] took three kinds of common apple leaf diseases as the research object, used an improved SVM classifier to identify apple leaf diseases, and finally realized an apple leaf disease recognition system. Shi et al. [4] achieved the recognition of three kinds of apple leaf diseases by using two-dimensional subspace learning dimension reduction (2DSLDR) based on differences in the shape, color, and texture of images of disease spots corresponding to different apple leaf diseases. Song et al. [5] proposed a one-to-one voting SVM strategy, which can effectively identify three kinds of apple leaf diseases, namely mosaic disease, rust disease, and Alternaria leaf spot disease. Wang et al. [6] applied a support vector machine optimized with a genetic algorithm for the recognition of apple mosaic disease, rust disease, and black spot disease and achieved a good recognition effect. Yu et al. [7] designed a two-layer structure model to identify apple leaf diseases.

Based on the above research, we found that traditional image processing methods mainly rely on expert experience to manually extract information on the color, texture, and shape of leaf disease images. Due to the complexity and diversity in the image background and disease spots of actually diseases leaves, the methods of artificial feature design and selection using traditional image processing technology are aimed at specific datasets. Therefore, the established disease recognition model is not universal, and the model migration ability is poor. However, in recent years, with the rise of deep learning techniques, CNNs (convolutional neural networks) have become a research focus for the automatic identification of agricultural plant diseases and insect pests. For example, Pallagani et al. [8] took 14 crop varieties and 26 diseases from the PlantVillage dataset as research objects, for which the recognition accuracy of the ResNet50 model reached 99.24%, and developed a dCrop app based on Android Studio. Albattah et al. [9] proposed an improved one-stage detector, CenterNet, with a 99.982% average accuracy on the PlantVillage dataset.

Some scholars focused their research on apple leaf diseases, as follows. Liu et al. [10] took four kinds of apple leaf diseases (leaf spot, rust, brown spot, and mosaic) as research objects and improved the original AlexNet network, with an overall accuracy of 97.62%. Baranwal et al. [11] proposed a new CNN based on LeNet-5 to identify black rot, scab, rust, and healthy apple leaves with an accuracy of 98.54%. Jiang et al. [12] proposed a CNN based on GoogLeNet’s Inception structure and Rainbow Connection structure, with which apple leaf diseases, including brown spot, gray spot, mosaic, and blotch, were identified, and the detection accuracy was 78.80%.

However, existing studies have only constructed a classification model for multiple single apple leaf diseases, and few researchers have considered the simultaneous occurrence of two diseases on a single leaf. In this case, when the symptoms of one disease are more pronounced than those of the other, the causes and symptoms of both diseases are very similar. Additionally, this similarity is the main challenge in establishing a classification model for apple leaf diseases [10,13].

The previously described methods do not fully cover these challenges, as they focus on late stages of the disease, cannot deal with diseases with similar symptoms, and do not support the simultaneous detection of different diseases on the same plant.

Apple trees are often affected by various diseases during their growth process; as such, many diseases may occur on the same leaf, and when the symptoms of one disease are more pronounced than those of other diseases, the diseases have very similar symptoms, which is the main challenge in building a multi-disease classification model for single apple leaves. Therefore, in this case, it is necessary to study a neural network model that can effectively extract and distinguish these fine-grained features.

There have been studies on multiple diseases on the same apple leaf, such as [14], which studied four types of apple leaf diseases in the Plant Pathology 2020 challenge dataset using the ResNet50 network model, with high recognition accuracy for three single-leaf and single-disease categories. However, the classification accuracy for the same leaf containing multiple disease symptoms was only 51%. Bansal et al. [15] proposed an ensemble of pre-trained DenseNet121, EfficientNetB7, and EfficientNet NoisyStudent that aims to classify four classes of diseased apple leaves (healthy, apple scab, apple cedar rust, and multiple diseases), achieving an accuracy of 96.25% on the test set but 90% accuracy for multiple diseases.

Therefore, in order to improve the accuracy of deep neural networks for multiple diseases appearing on the same apple leaf, an optimized RegNet [16] network was proposed. This study took seven common apple leaf diseases as the research objects, including healthy leaves, rust leaves, scab leaves, ring rot leaves, *Panonychus ulmi* disease leaves, both rust and scab diseases leaves, and both *Panonychus ulmi* symptomatic disease and ring rot disease leaves, using the optimized RegNet model to identify the small-sample-size apple leaf disease dataset collected in this study.

## 2. Materials and Methods

### 2.1. Image Datasets

The dataset used in this study consists of two parts. Images in the first part were taken by members of the research group using mobile phones during May 2020 and September 2021 at the planting base and farmer orchards of the Fruit Tree Institute of Shanxi Academy of Agricultural Sciences. In order to include as many environmental factors related to plant growth as possible in the collected data and meet the diversity requirements of data sampling, the data were collected during three time periods, namely morning, noon, and evening, under both sunny and cloudy weather conditions. The 7 collected apple leaf image samples (healthy + 6 diseases) included the entire growth period from apple tree germination to harvesting, and each disease image sample contained different stages of onset.

Considering the insufficient number of collected apple leaf disease images and the fact that the background and category of the apple disease leaf images in the “Plant Pathology Challenge” for CVPR 2020-FGVC7 (https://www.kaggle.com/c/plantpathology-2020 fgvc7, accessed on 20 December 2023) [17] are consistent with the collected image data, the two were fused to form the dataset used in this study. Additionally, we collaborated with professional scholars to screen, classify, and organize the above two image types one by one, establishing a relatively reliable dataset of apple leaf diseases. A total of 2609 images of apple leaf diseases were obtained, among which the numbers of images of healthy, scab, rust, *Panonychus ulmi* symptoms, ring rot, both *Panonychus ulmi* symptoms and ring rot, and both apple scab and rust were 738, 647, 710, 106, 141, 176, and 91, respectively. *Panonychus ulmi* symptoms refer to the presence of lesions caused by mites.

It is worth noting that this dataset includes diseases that produce very similar symptoms, especially in the early stages, and there may be more than one disease on the same leaf. The fact that they exhibit similar visual symptoms increases the complexity of the dataset. Because it is a highly complex dataset, where different diseases have similar or slightly different symptoms, it is difficult to use universal algorithms for processing, especially of early symptoms. Furthermore, all images in the dataset include complex background fields, which ensures that the studied methods offer high generalization performance and robustness. Examples of diseased apple leaf samples are shown in Figure 1.

In order to study the effect of the image background on model performance, two datasets were used in this study. Dataset 1 consists of the original image taken in the field (i.e., the image contains a large area of background information other than the main leaf). Dataset 2 was obtained by cropping the surrounding background with a minimum bounding rectangle centered on the target main leaf (i.e., containing a small amount of background information). Examples of images in Dataset 1 and Dataset 2 are shown in Figure 2.

#### Data Augmentation

The original image dataset used in this study is small, and there is an imbalance in the number of species, while the CNN models are complex and easily result in overfitting. Therefore, it is necessary to expand the collected dataset to achieve species balance, enrich data, increase the number and diversity of samples, improve the robustness, and avoid overfitting of the model [18].

There are two kinds of data enhancement methods, namely online enhancement and offline enhancement. Offline enhancement deals with the dataset directly, wherein the number of data becomes the enhancement factor multiplied by the number of samples in the original dataset. This method is often used when the dataset is small. The online enhancement method is mainly used to obtain batch data, then enhance the batch data (including image rotation, Gaussian noise, and random erasure) and is often used for larger datasets. In addition, many machine learning frameworks have already supported this kind of data enhancement, and the online data enhancement process proposed in this paper is realized by using the Image Data Generator module in Keras.

First of all, aiming at the problem of imbalance between data classes in this study, we ensure that each class has about 700 images by using offline data enhancement such as image rotation, Gaussian noise, and random erasure. Then, the balanced dataset is randomly divided into 10 pieces, one of which is selected as the test set, while the remaining data are designated as the training set and validation set.

Secondly, in view of the problem of there being a limited amount of data in this study, and to prevent overfitting of the model, the original training set and the validation set are expanded to three times the original size by offline enhancement, online enhancement, and offline enhancement followed online enhancement. Finally, six different expanded datasets are obtained, namely dataset1+A, dataset1+B, dataset2+A, dataset2+B, dataset1+A+B, and dataset2+A+B. Here, dataset1+A refers to the offline expansion dataset used by Dataset 1, dataset1+B refers to the online expansion dataset used by Dataset 1, and dataset1+A+B refers to the offline expansion and then online expansion dataset used by Dataset 1. The data distributions of the datasets are shown in Table 1, where the augmented dataset is the sum of the original training set and the validation set, which were expanded three times.

### 2.2. Construction of Model for Identifying Apple Leaf Diseases

#### 2.2.1. Construction of RegNet

In this study, an optimized lightweight identification framework for multiple and unidentifiable diseases of apple leaves is proposed.

The core aim of the RegNet [19] model is not to design any particular network or even to find a particular kind of network but to search the design space of the network. The design space was progressively designed. In the design process, the input is the initial design space, and the output is a simpler and more effective design space, which is progressively iterated to optimize the design space. According to the design space, the design criteria of the model are abstracted so that they can be transferred to different hardware environments and the network details can be flexibly adjusted on the basis of these design criteria according to different environments.

The design space refers to the set of concrete architectures instantiated from the model family, the parameterization of the model family, and the set of values allowed for each hyperparameter. By fixing the value of the hyperparameter in the design space, a concrete model can be obtained. The structure of the RegNet model is shown in Figure 3.

(a) As shown in Figure 3a, the network consists of a stem, followed by the network body that performs most of the calculations and a head. Among them, the stem is a regular convolution layer (default includes batch normalization and ReLU activation function), with a kernel size of 3 × 3, a stride of 2, and 32 convolution kernels. The head commonly a classifier in classification networks, consisting of a global average pooling layer and a fully connected layer.

(b) As shown in Figure 3b, the body consists of a series of stages that run at a gradually decreasing resolution.

(c) As shown in Figure 3c, each stage is composed of a series of identical blocks, and except for the first block, which uses a convolution with a stride of 2, all other blocks use a convolution with a stride of 1. The construction of a block with stride = 1 and stride = 2 is shown in Figure 4a,b, respectively.

Each block consists of a 1 × 1 convolution, a 3 × 3 group convolution, and a 1 × 1 convolution, where the 1 × 1 convolution changes the channel width. Additionally, each convolution is followed by BN+ReLU, where r represents resolution, w represents the channel width of the feature matrix, b represents bottleneck ratio, and g represents the group width.

When stride = 1, the shortcut does not perform any processing. When stride = 2, the shortcut branch down-samples through a 1 × 1 convolution.

#### 2.2.2. Optimization of RegNet Model

Compared with VGG16, GoogleNet, and other networks, RegNet has the advantages of a short training time, lightweight model files, and good convergence in crop disease recognition, making it easy to deploy to mobile devices [20]. Therefore, this study chose RegNet as the identification model, and to solve the problems of weak generalization ability and large model volume, the RegNet network structure was optimized.

First, the ReLU6 activation function is used to replace the original Sigmoid activation function to improve fully connected layer 2 (FC2). At the same time, the dropout layer and the improved Bottleneck operator are introduced to improve computational speed and facilitate the deployment of the model to mobile devices. Secondly, the Ranger optimizer is used to replace the original SGD optimizer [21]. Finally, transfer learning is adopted to load the trained weight parameters into the convolution layer as the initialization weight parameters of the model, improving the generalization ability of the model [22].

The ReLU function is defined by Equation (1) as follows:(1)$$f(x)=\max\left(0,x\right)$$

#### 2.2.3. Ranger

Ranger [23] is an optimizer proposed by Tong et al. in 2019 that combines RAdam [23] and LookAhead [24]. The function of RAdam is to use a dynamic rectifier to adjust the adaptive momentum of Adam based on variance and effectively provide automatic warm-up to ensure a good start to training based on the current dataset. Inspired by the latest advances in deep neural network loss surface understanding, LookAhead retains an additional weight copy, then allows the internalized “faster” optimizer (i.e., RAdam) to perform a search of five or six batches (the batch intervals specified by the k parameter). When the exploration of k batches is completed, LookAhead multiplies the difference between its saved weight and the latest weight of RAdam by an alpha parameter (0.5 by default), then updates the weight of RAdam. This allows for “foresight” or exploration of faster weight sets, while leaving slower weight sets behind to maintain long-term stability. The result is faster convergence for different deep learning tasks with minimal computational overhead while reducing the need for extensive hyperparameter tuning.

#### 2.2.4. Experimental Setup

The experimental environment was Ubuntu 18.04 running on an Intel Core i9 9820X with 64G RAM, a GeForce RTX 2080Ti 11G DDR6, and TensorFlow and Cuda10.1, which were used for training.

To avoid memory overflow, the batch size used for both the validation and training sets was 16, the number of iterations was 50, the dropout was 0.5, and the cross-entropy loss function in Keras was used. For the model to converge better in terms of recognition accuracy, the experiments used a learning-rate decay strategy, with the initial learning rate set to 0.005 and the learning rate decaying to 90% of the original rate after every five epochs.

The experiments consist of the following four scenarios: (1) We studied the effects of different data enhancement methods on the model’s performance. (2) Based on the baseline network ResNet50 model [25] and a series of improved models (such as ResNeXt [26] and ResNeSt [27]), the performance of ResNet50 on different image background datasets was compared and analyzed. (3) We studied the effects of using different training methods and different optimizers on the performance of the model based on ResNet50 and its series of improved models. (4) Finally, the generalization performance of RegNet was analyzed.

## 3. Results and Analysis

### 3.1. Effect of Data Enhancement on Model Performance

Data enhancement technology can generate more data from limited data; increase the number, as well as the diversity, of samples; and, thus, improve the generalization ability of the model. In view of the problem of there being a small amount of data in this study, in order to avoid the overfitting phenomenon, a data enhancement operation is needed. The purpose of this section is to investigate the effect of different data enhancement methods on the generalization performance of the model.

To study the effects of different image enhancement methods on the performance of the model, the baseline network ResNet50 model was used to study the influence of three different data enhancement methods (offline augmentation, online augmentation, and offline augmentation before online augmentation) on the generalization performance.

Figure 5 shows the validation accuracy and loss function curves of the baseline network ResNet50 model on Dataset 1 and Dataset 2 using the same training method (transfer learning training for all layers) in six sets of experiments with different data enhancement methods.

In the figure, dataset1 and dataset2 represent the original datasets without any data enhancement methods; dataset1+A and dataset2+A are datasets expanded by offline data enhancement for Dataset 1 and Dataset 2, respectively; dataset1+B and dataset2+B are datasets expanded by online data enhancement for Dataset 1 and Dataset 2, respectively; and dataset1+A+B and dataset2+A+B represent Dataset 1 and Dataset 2 enhanced and expanded with offline and online data augmentation, respectively. The solid line represents the accuracy curve and loss function curve of ResNet50 on Dataset 1, and the line with data marker points represents the accuracy curve and loss function curve of ResNet50 on Dataset 2.

Overall, it is clear from Figure 5 that some augmentation methods a greater impact on the performance of the model on the same dataset. For example, for Dataset 1, using offline augmentation method proved to be much better than using the other two augmentation methods, the effects of which are not very different. For Dataset 2, the validation accuracy of the three different methods of augmentation varied greatly, and offline augmentation was the best, while online augmentation was the worst. It can be concluded that the model can achieve better classification performance by only using offline augmentation rather than the other two augmentation methods. This may be because the online dynamic data expansion method saves the significant amount of space needed to store the expanded data, enriches data diversity, and can reduce the overfitting phenomenon of the model; however, to some extent, the sample distribution of the original dataset is destroyed, and the training volatility is increased.

### 3.2. Effect of Image Background on Model Performance

This section aims to investigate the impact of different image backgrounds on model performance. Due to the small amount of data in the apple leaf disease dataset used in this study, in order to avoid overfitting, Dataset 1 and Dataset 2 were expanded using offline enhancement. This section describes the use of the RegNet and ResNet50 models and a series of improved models (such as ResNeXt50 and ResNeSt), as well as the analysis of the impacts of different image backgrounds on model performance. Figure 6a,b show the validation accuracy and loss function curves on different datasets, respectively. The solid line represents the accuracy curve and loss function curve of each model on Dataset 1, and the line with data marker points represents the accuracy curve and loss function curve of each model on Dataset 2.

By observing the accuracy curves of two different datasets (that is, two curves of the same color), it can be concluded that the performance of the four models on Dataset 1 is better than that on Dataset 2, as seen in Figure 6a. In addition, the accuracy curves on Dataset 2 converge faster and fluctuate less.

In summary, by comparing the two datasets with the same data enhancement method, it was found that the model using Dataset 2 was better than that using Dataset 1, which may be because the captured apple leaf disease images contain too much background information that is irrelevant to the target, and the clipping operation could have reduced the complexity of the image background. Therefore, the effects of different training methods and optimizers on model performance will be investigated based on Dataset 2 in subsequent work.

### 3.3. Comparison of Different Training Methods

There are three strategies for training. One is to initialize the model parameters randomly in the constructed model (training from scratch), and the second is to initialize the weights of the model using the weight parameters pre-trained on ImageNet and fine-tune all layers of the network structure (i.e., transfer learning strategy one). The third is to initialize the weights of the model using weights pre-trained on ImageNet, then only fine-tune the weights of fully connected layers (i.e., transfer learning strategy two). In this section, all models use the Adam optimizer are investigated. Figure 7a–d show the accuracy curves of four network models on Dataset 2 using different training methods. In the figure, the black solid line represents transfer learning strategy one, represented by T1. Green represents transfer learning strategy two, represented by T2. The blue line represents the learning strategy trained from scratch, represented by T0.

In Figure 7, it can be seen that there are three points to consider. First, according to the analysis of convergence speed, all four network models trained from scratch have the slowest convergence speed and exhibit significant fluctuations. The convergence speed using transfer learning strategy one is faster than that of training from scratch. Additionally, the convergence speed of transfer learning strategy two is the fastest, reaching a stable convergence state after only a dozen rounds of training. This is because transfer learning strategy two has fewer trainable parameters than transfer learning strategy one. Secondly, from the accuracy curve, it can be seen that for all models, transfer learning strategy one can achieve higher classification accuracy than transfer learning strategy two. The reason for this may be that, although the convolution modules trained on the ImageNet dataset can be used to extract image features, there are significant differences (quantity, type, and size) between the two datasets compared to the dataset used in this study. The use of transfer learning strategy two cannot achieve ideal results, while the testing accuracies of the four network models using transfer learning strategy one are significantly improved. This indicates that training only the classification layer cannot make the model adapt well to the data used in this study.

In summary, transfer learning can accelerate the convergence of the network and improve the accuracy of training and testing. Therefore, owing to the small sample size of the apple leaf disease dataset, it is necessary to adopt the training method of transfer learning to fine-tune all layer parameters.

### 3.4. Comparison of Different Optimizers

Classical deep learning architectures such as ResNet, ResNeXt, ResNeSt, and RegNet are trained using typical SGD optimizers, so we attempted to use other optimizers, such as Adam, RAdam, and Ranger, to compare the performance of the models.

(1) Comparison of the performance of the same model under different optimizers

Figure 8a–d illustrate the accuracy curves of four models, namely RegNet, ResNeSt, ResNet, and ResNeXt, using different optimizers on the validation set.

In Figure 8a and Table 2, it can be seen that the final convergence states of the RegNet network model using the four optimizers were not significantly different, except for the starting point of the model and the convergence speed. The Ranger optimizer had the highest starting point and the fastest convergence speed, and the SGD optimizer had the second fastest convergence speed after Ranger, while the Adam optimizer had the lowest starting point and the slowest convergence speed.

The ResNeSt network model achieved the worst performance using the SGD optimizer model, with the lowest starting point of the accuracy curve and the lowest convergence state. The model’s accuracy curve fluctuates the most under the RAdam optimizer, and the maximum accuracy of the model was the lowest, as presented in Figure 8b. However, the effect difference between the Adam and Ranger optimizers was small, and the final convergence state of the model under the Ranger optimizer was better than that under the Adam optimizer. Therefore, the performance of the model was influenced by the optimizer. In conclusion, the Ranger optimizer is the best-suited optimizer for the ResNeSt network model.

According to Figure 8c,d and Table 2, the use of the Ranger optimizer had the highest starting point, fastest convergence speed, and highest accuracy on the test set for the ResNet and ResNeXt models.

In summary, of the four models, SGD achieved the worst performance; RAdam had the largest fluctuation; and Ranger had the best effect, the highest accuracy, the most stable convergence, and the fastest convergence. This shows that Ranger has the best universality.

(2) Comparison of performance of different models under the same optimizer

Figure 9 illustrates the accuracy curves of the RegNet, ResNeSt, ResNet, and ResNeXt models using Ranger on the validation set.

Figure 9 illustrates that, with other models, RegNet had the fastest convergence speed and achieved good convergence in the 16th round, while other models mostly started to achieve good convergence in the 25th to 30th rounds. Additionally, the RegNet network model had a higher convergence starting point, faster convergence speed, and higher accuracy than the other three models. The accuracy curves of the other three network models showed little difference, and with the ResNet network model, there was a relatively large oscillation in the accuracy curve. According to the above analysis, the recognition accuracy and convergence of the RegNet model on the apple leaf dataset were better than those of the other models. Therefore, the well-performing RegNet network model equipped with a universal Ranger optimizer will be selected for subsequent studies.

### 3.5. Test of Model Generalization Performance

In order to verify the generalization performance of the RegNet model, a comparative analysis was conducted on the test results of the models trained on Datasets 1 and 2. Additionally, four confusion matrixes were obtained, as presented in Figure 10a–d.

The confusion matrix seen in Figure 10a (dataset1-dataset1) represents the results trained on dataset1 and tested on dataset1, Figure 10b (dataset1-dataset2) represents the results obtained by the model trained on Dataset 1 and tested on Dataset 2, Figure 10c (dataset2-dataset1) represents the results obtained by the model trained on Dataset 2 and tested on Dataset 1, and Figure 10d (dataset2-dataset2) represents the results obtained by the model trained on Dataset 2 and tested on Dataset 2. The main diagonal numbers are the numbers of correctly predicted sample images, and the numbers in other positions correspond to the numbers of incorrectly predicted sample images. Based on the confusion matrix, the generalization performance of the models was evaluated using four indicators, namely precision, recall, specificity, and accuracy. The specific results are detailed in Table 3.

As shown in Figure 10a, the recognition accuracy of the model tested on Dataset 1 was 99.74%, and only one sample was misclassified, namely the sample for which a leaf with both scab and rust was misclassified as rust. The overall recognition accuracy of the model on Dataset 2 was 90.77%, and a total of 36 samples were misclassified, which is much lower than the accuracy of the model tested on Dataset 1 (99.74%).

In Figure 10, the confusion matrix (d) shows that the overall recognition accuracy of the model is 99.23%. Except for three samples that were misclassified, all other samples were correctly identified. One sample with *Panonychus ulmi* symptoms was misclassified as *Panonychus ulmi* symptoms and ring rot, one sample with both *Panonychus ulmi* symptom and ring rot was misclassified as ring rot, and another sample with ring spot disease was misclassified as having both *Panonychus ulmi* symptoms and ring rot. As can be seen from the confusion matrix (c), the overall recognition accuracy of the model tested on Dataset 1 was only 93.85%, with a total of 24 samples misclassified, which is far lower than the accuracy of the model tested on Dataset 2 (99.23%).

In fact, the test accuracy of the model trained on different datasets is above 90%, which shows that the model achieved good generalization performance. After careful analysis, it was found that the model trained on Dataset 2 had a good prediction ability for both Dataset 1 and Dataset 2. Therefore, the model trained on Dataset 2 has a stronger generalization ability and can be better applied in actual production. The same conclusion can be drawn from the evaluation indicators (precision, recall, specificity, and accuracy) listed in Table 3. 

## 4. Discussion

This paper proposes a lightweight RegNet model for optimized for operation in complex environments through a series of improvements. The model effectively solves the problem of multiple diseases on a single leaf seen in images with complex backgrounds.

The following conclusions are drawn from the experimental results. First, the two transfer learning training methods can significantly accelerate network convergence and improve classification performance. Second, background cropping can improve the model’s extraction of disease feature information. Finally, the models trained with Ranger were more robust and accurate.

More precisely, the optimized RegNet model achieves better disease identification for images with complex backgrounds, with an accuracy of 99.23% on the test set and 99.1% accuracy on images showing multiple diseases. In addition, our model achieves significant improvement in identifying multiple diseases on a single leaf compared with the models proposed in references [15,17], which reported accuracies of 51% and 90, respectively. The main reason classification errors occur when two diseases can be found on the same leaf and when two diseases are found separately is that the symptoms of one disease may be very similar when the symptoms of other diseases are more pronounced. This similarity is the main challenge associated with the proposed classification model when considering single apple leaves and multiple diseases. 

Although our model achieves better recognition of multiple diseases on single leaves in complex backgrounds, there are some undeniable shortcomings identified in this study in terms of data acquisition and processing. First, the model had strong uncertainty and poor interpretability due to the insufficient diversity in the disease dataset (such as early symptoms of diseases). Secondly, the performance of the model on other plant disease datasets was not verified. Finally, there is still room for research on applications embedded in mobile devices. Therefore, it is hoped that in the future, more in-depth research will be conducted on the above-mentioned issues using existing research methods.

## 5. Conclusions

In order to solve the problem of feature extraction being time-consuming and laborious when using traditional recognition methods, this paper studied the images of seven kinds of diseased apple leaves (including situations wherein multiple diseases occur on the same leaf) by using an improved RegNet depth convolution neural network model. The effects of training mode, data expansion methods, optimizer selection, image background, and other factors were compared and analyzed. The conclusions are as follows:

(1) It was concluded that using only offline augmentation is better than online expansion and a combination of offline and online expansion.

(2) The effects of training methods, image backgrounds, and the choice of optimizer on the performance of four classification models, namely ResNet50, ResNeXt50, ResNeSt, and RegNet, were studied. The experimental results show that, compared to training from scratch, the two transfer learning training methods can significantly accelerate network convergence, improve classification performance, and shorten the training time of the model. Secondly, background cropping can, to some extent, eliminate background information, improve the model’s performance in extracting disease feature information, thereby improving the overall performance of the model. Finally, the four models trained with the Ranger optimizer were more stable and accurate, and RegNet with Ranger performed the best.

(3) In order to analyze the generalization performance of the RegNet network model, it was trained on Dataset 1 and Dataset 2 and tested on both datasets separately. The results show that the model trained on Dataset 2 had a good predictive ability for both Dataset 1 and Dataset 2, and it was concluded that the model trained on dataset2 had a better generalization ability.

In summary, our work supports the identification of multiple diseases on the same apple leaf under complex field backgrounds. Additionally, the results show that it has the potential to accurately identify apple diseases, which will help orchard managers save manpower input and reduce pesticide use.

## Figures and Tables

**Figure 1 plants-13-01535-f001:**
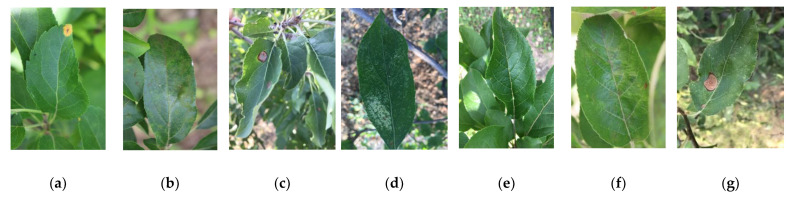
Examples of different disease images: (**a**) rust; (**b**) scab; (**c**) ring rot; (**d**) *Panonychus ulmi* symptoms; (**e**) healthy; (**f**) rust + scab; (**g**) *Panonychus ulmi* symptoms + ring rot.

**Figure 2 plants-13-01535-f002:**
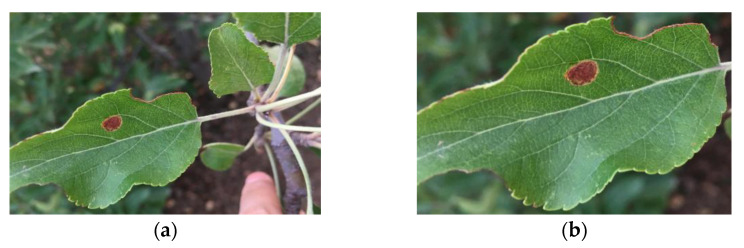
Examples from the two datasets: (**a**) example of leaf in Dataset 1; (**b**) example of leaf in Dataset 2.

**Figure 3 plants-13-01535-f003:**
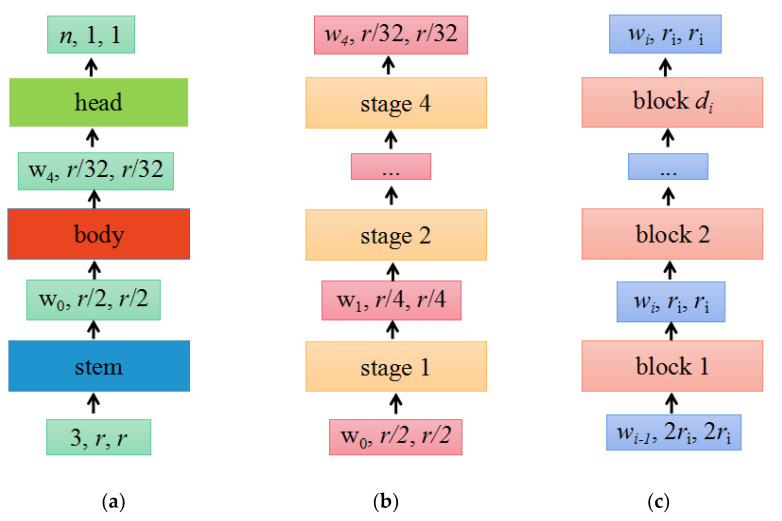
The network mainly consists of three parts, namely the stem, body, and head. (**a**) Network; (**b**) body; (**c**) stage i.

**Figure 4 plants-13-01535-f004:**
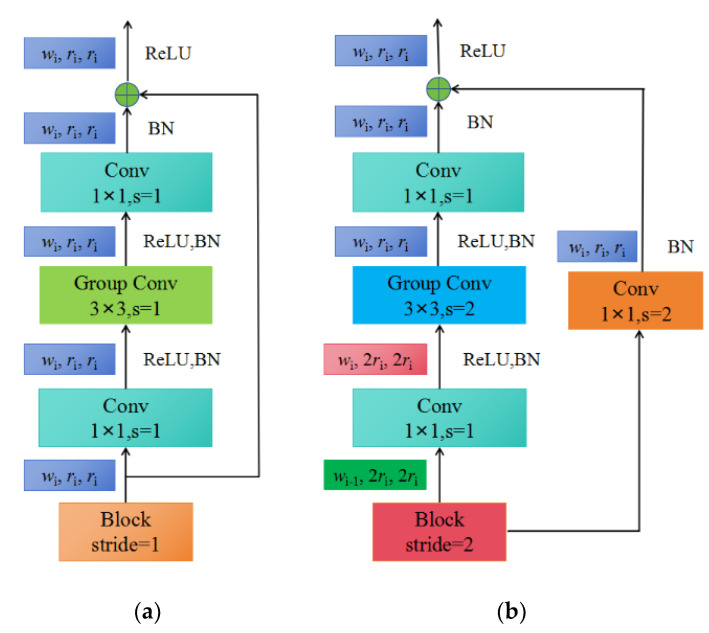
The construction of blocks: (**a**) X block, stride = 1; (**b**) X block, stride = 2.

**Figure 5 plants-13-01535-f005:**
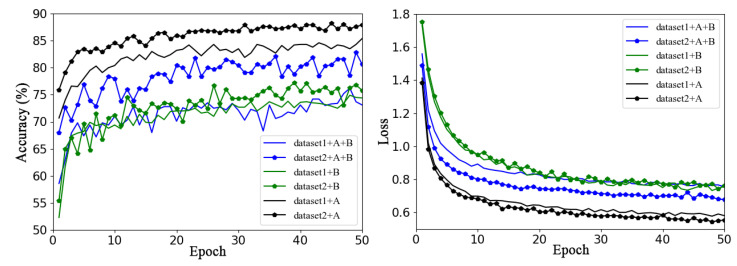
Performance of different augmentation methods on Dataset 1 and Dataset 2, where A refers offline enhancement and B refers online enhancement.

**Figure 6 plants-13-01535-f006:**
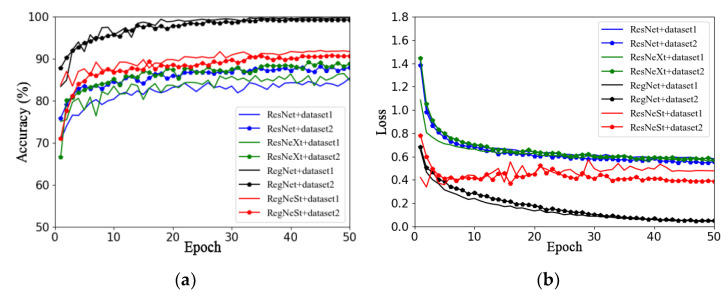
Performance comparison of models on different datasets: (**a**) accuracy; (**b**) loss.

**Figure 7 plants-13-01535-f007:**
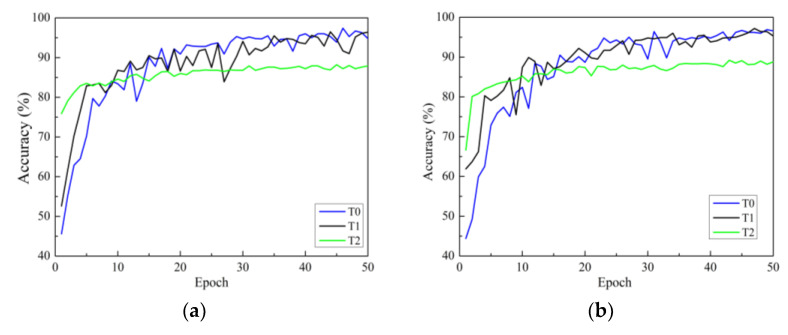
Validation accuracy curves under different training methods: (**a**) ResNet; (**b**) ResNeXt; (**c**) ResNeSt; (**d**) RegNet. T0 is the model trained from scratch, T1 refers to transfer learning strategy one, and T2 refers to transfer learning strategy two.

**Figure 8 plants-13-01535-f008:**
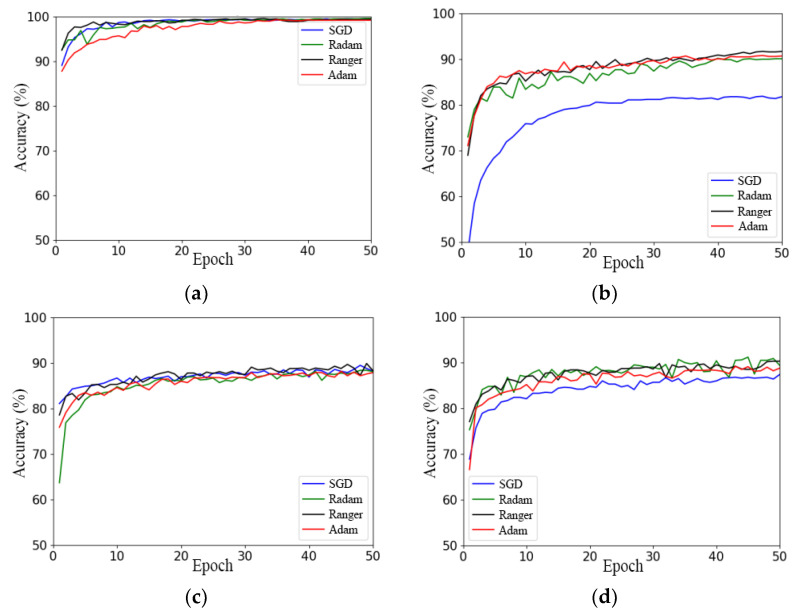
Comparison of different optimizers used with the models: (**a**) RegNet; (**b**) ResNeSt; (**c**) ResNet; (**d**) ResNeXt.

**Figure 9 plants-13-01535-f009:**
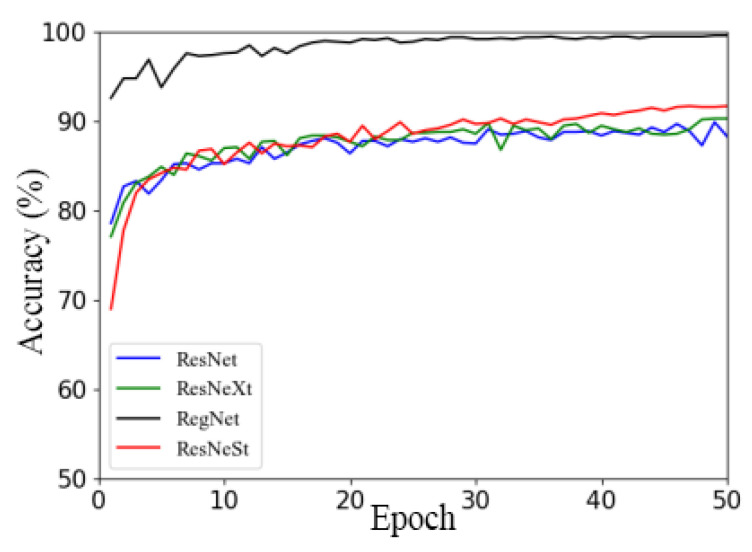
Comparison of different models under Ranger optimizer.

**Figure 10 plants-13-01535-f010:**
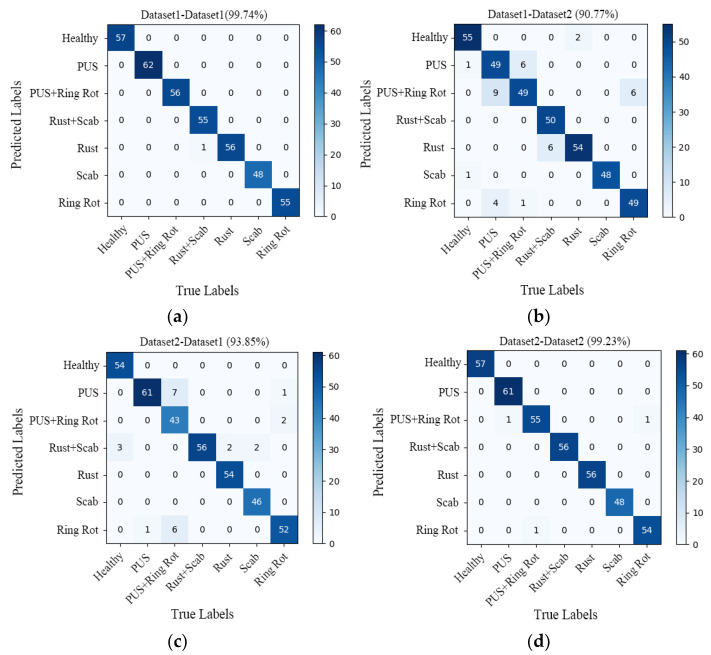
Confusion matrix of (**a**) dataset1-dataset1 (99.74%); (**b**) dataset1-dataset2 (90.77%); (**c**) dataset2-dataset1 (93.85%); (**d**) dataset2-dataset2 (99.23%). PUS refers *Panonychus ulmi* symptoms.

**Table 1 plants-13-01535-t001:** The distributions of the datasets (numbers refer to number of images).

Classes	Original Dataset	Balanced Dataset	Training + Validation Set	Augmented Dataset	Test Set
Healthy	738	738	534+147	2043	57
Scab	647	647	470+129	1797	48
Rust	710	710	512+142	1962	56
Ring Rot	141	705	509+141	1950	55
*Panonychus ulmi* symptom	106	742	532+148	2040	62
Scab + rust	91	728	527+145	2016	56
Ring rot + *Panonychus ulmi* symptoms	176	704	508+140	1944	56
Total	2609	4974	3592+992	13,752	390

**Table 2 plants-13-01535-t002:** Comparison of model test accuracy.

Optimizer	Models			
	RegNet	ResNeSt	ResNet	ResNeXt
SGD	99.5	81.9	89.4	87.4
Ranger	99.6	91.7	89.5	91.2
RAdam	99.6	90.2	88.4	90.3
Adam	99.3	90.8	88.2	89.2

**Table 3 plants-13-01535-t003:** The results of different models. PUS refers *Panonychus ulmi* symptoms.

Training Set-Test Set	Category	Precision	Recall	Specificity	Accuracy (%)
Dataset 1-Dataset 1	Healthy	1	1	1	99.74%
	PUS	1	1	1	
	PUS + Ring Rot	1	1	1	
	Rust + Scab	1	0.982	1	
	Rust	0.982	1	0.997	
	Scab	1	1	1	
	Ring Rot	1	1	1	
Dataset 1-Dataset 2	Healthy	0.977	0.921	0.997	90.77%
	PUS	0.891	0.745	0.983	
	PUS + Ring Rot	0.768	0.899	0.952	
	Rust + Scab	0.973	0.957	0.994	
	Rust	0.93	0.978	0.989	
	Scab	0.97	0.963	0.996	
	Ring Rot	0.882	0.914	0.981	
Dataset 2-Dataset 1	Healthy	1	0.947	1	93.85%
	PUS	0.884	0.984	0.976	
	PUS + Ring Rot	0.956	0.768	0.994	
	Rust + Scab	0.889	1	0.979	
	Rust	1	0.964	1	
	Scab	1	0.958	1	
	Ring Rot	0.881	0.945	0.979	
Dataset 2-Dataset 2	Healthy	1	1	1	99.23%
	PUS	1	0.99	1	
	PUS + Ring Rot	0.994	0.994	0.999	
	Rust + Scab	1	1	1	
	Rust	1	1	1	
	Scab	1	1	1	
	Ring Rot	0.993	1	0.999	

## Data Availability

Data are contained within the article.

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
