# Peer review of "Identification of Multiple Diseases in Apple Leaf Based on Optimized Lightweight Convolutional Neural Network"

_plants, 2024, doi:10.3390/plants13111535_

Round 1

Reviewer 1 Report

Comments and Suggestions for Authors

The authors of this manuscript present the use of transfer learning for the detection of seven apple leaf diseases, including mixed infections. The manuscript has several flaws which have to be addressed before I would reconsider it. Given the subject matter, I also urge the authors to consider submitting to a better suited journal, such as Complex & Intelligent Systems or similar.

Numerous studies have used Deep learning to classify more than 7 plant diseases (e.g., Albattah et al. 2022 https://doi.org/10.1007/s40747-021-00536-1, Pallagani et al. 2019 10.1109/iSES47678.2019.00020). This manuscript should be placed into the context of this type of research, both in the Introduction and Discussion sections.

Considering that the main audience of Plants aren't machine learning experts, an additional figure describing the full analysis pipeline would help in understanding the methods. Methods should also be described accordingly, with the audience in mind.

Dataset1 and dataset2 refer only to the data acquisition described in Lines 98-107?

Clearly describe how and why the Plant pathology challenge data were used.

How were the augmentation techniques chosen? Have you checked for bias, i.e., that the algorithm has not learned the transformation as a feature of the images?

Have you tested the models on an unaugmented dataset, to ensure generalization?

Lines 159-161: So the train/test split was 90/10? What does "piece" mean here? How was the split into training and validation performed? Did you use stratified splits? If augmented versions of an image appear in both the training and test sets, the model may not be learning to generalize from new data but rather memorizing specific augmented features that it has already seen during training. This can lead to overfitting. Did you check your results for overfitting?

Table 1: What are the numbers? I expect them to be the number of images. What is the Augmented dataset? Shouldn't it be a sum of training+validation+test?

The Results section is a mix of methods, results, and discussion. Has to be corrected and improved. For example, Lines 234-245 belong to methods, Lines 438-448 to Discussion. Several algorithms are mentioned in the Results (e.g., optimizers), but are not mentioned in the Methods section.

Figures should be stand-alone, a reader should be able to understand that figure wihout having to search the text for descriptions.

In all figures of performance of augmentation methods, what is "value"? Remove the title for each chart ("Accuracy" and "Loss"), and use these titles as y-axis labels.

Table 2: Include other performance metrics, not just accuracy. F1, precision, recall, AUC...

Figure 11: x-axis labels are missaligned and overlap. Translate labels to English ("tongxin", "huaban"...).

The discussion section barely discusses the findings. Has to be improved.

Comments on the Quality of English Language

The manuscript requires additional English editing and proofreading. Some sentences are difficult to understand or are missing words (e.g., Lines 89-90).

Panonchyus ulmi is the latin name and should be in italics, and the genera name with the first letter in capital.

Author Response

Dear Reviewer,   We appreciate the reviewer for the comment. Please see the attachment.

Reviewer 2 Report

Comments and Suggestions for Authors

This study proposes the use of an optimized RegNet model to identify seven common apple leaf diseases. The study conducted comparisons and analyses on the impact of various factors such as training methods, data expansion
methods, optimizer selection, image background, and other factors on model performance. The authors have correctly reported the state-of-the-art literature on the problem focusing also on the additional problems of multiple single apple leaf diseases, and  the simultaneous occurrence of two diseases on a single leaf. In the Section 2 the authors describe the dataset used and some data enhancement methods in order to increase the size and to balance the datasets. Moreover they quite precisely describe the different CNNs used for testing and the experimental setup. In the Section 3 they extensively report the experimental results obtained by applying different strategies on model performances as for examples the data enhancement strategy, the image background, the training methods and several optimizers. Finally in the Discussion and Conclusion sections the authors explain the ratio beyond the pipeline implemented and teh relative findings obtained from the results. The paper is accepted in present form

Comments on the Quality of English Language

The quality of English language is quite good, although there are several punctuation errors

Author Response

Dear Reviewer,   We appreciate the reviewer for the comment. We have corrected punctuation errors in the manuscript.

Reviewer 3 Report

Comments and Suggestions for Authors

This paper is very important nowadays and is of great interest in agriculture for the prevention and control of the development of food, more specifically apples. A very well written, organized and explained paper. In particular, the description and explanations of the methods applied are detailed and clear. I really liked it and recommend it for publication. I can only add two small suggestions:

- line 150, extra white space.

- the caption for figure 9 must come with the figure.

Author Response

(The authors gave the same response as above.)

Round 2

Reviewer 1 Report

Comments and Suggestions for Authors

Thank you for the revised manuscript. It is much better and easier to read and understand. The results are also clearer. I have one minor and one major comment.

Comment 1:

Revised Table 3: Add a horizontal line to divide the training set-test set combinations. It will make the table a bit easier to read.

Comment 2:

The discussion section is still much too weak. Parts of results still read like a discussion. For example, lines 358 - 377 are a mix of results and discussion. You should interpret your findings, discuss the implications of your methods and results, and place your research into a wider context. What are the main contributions of your research, what are the breakthroughs, how do they compare to previous research? Address the limitations of existing methods and explain how your approach overcomes them. Discuss the robustness of your model across different data sets and plant species, as this is the most important aspect for implementation. Would your method be useful for early, pre-symptomatic detection? Would it be any better than existing reserach? What about interpretability of your models, is it an issue? Could you use explainable AI approaches to improve your research and make the models more interpretable? Which were the main challenges you faced? What should be future research directions, which are current knowledge gaps? How can end-users (farmers, agronomists, policy-makers) benefit from your research?

Comments on the Quality of English Language

General comment; there are a few minor mistakes, e.g. missing full stop on line 417, and full stop instead of comma on line 445. I recommend you carefully re-read the manuscript and look for such minor mistakes.

Author Response

(The authors gave the same response as above.)
